# Hydrogen-Rich Alkaline Water Supplementation Restores a Healthy State and Redox Balance in H_2_O_2_-Treated Mice

**DOI:** 10.3390/ijms25126736

**Published:** 2024-06-19

**Authors:** Davide Mizzoni, Mariantonia Logozzi, Rossella Di Raimo, Massimo Spada, Stefano Fais

**Affiliations:** 1Exo Lab Italia, Tecnopolo d’Abruzzo, Strada Statale 17, Località Boschetto di Pile, 67100 L’Aquila, Italy; davide@exolabitalia.com (D.M.); rossella@exolabitalia.com (R.D.R.); 2Department of Oncology and Molecular Medicine, Istituto Superiore di Sanità, Viale Regina Elena 299, 00161 Rome, Italy; mariantonia.logozzi@iss.it; 3Department of Centro Nazionale Sperimentazione e Benessere Animale, Istituto Superiore di Sanità, Viale Regina Elena 299, 00161 Rome, Italy; massimo.spada@iss.it

**Keywords:** hydrogen-rich alkaline water, redox balance, antioxidant, ROS, glutatione, mitochondrial membrane potential, antiaging, superoxide, DNA damage

## Abstract

Water is a major requirement for our bodies, and alkaline water has induced an antioxidant response in a model of natural aging. A series of recent reports have shown that aging is related to reduced water intake. Hydrogen-rich water has been suggested to exert a general antioxidant effect in relation to both improving lifestyle and preventing a series of diseases. Here, we wanted to investigate the effect of the daily intake of hydrogen-rich alkaline water (HAW) in counteracting the redox imbalance induced in a model of H_2_O_2_-treated mice. Mice were treated with H_2_O_2_ for two weeks and either left untreated or supplied with HAW. The results show that HAW induced a reduction in the ROS plasmatic levels that was consistent with the increase in the circulating glutathione. At the same time, the reduction in plasmatic 8-hydroxy-2′-deoxyguanosine was associated with reduced DNA damage in the whole body. Further analysis of the spleen and bone marrow cells showed a reduced ROS content consistent with a significantly reduced mitochondrial membrane potential and superoxide accumulation and an increase in spontaneous proliferation. This study provides evidence for a clear preventive and curative effect of HAW in a condition of systemic toxic condition and redox imbalance.

## 1. Introduction

Preventing and curing the redox imbalance that is associated with aging is a central issue in health care [1,2,3]. A general strategy is to use antioxidant molecules that may help in maintaining a balance between reduction and oxidation. We recently published data showing that alkaline water supplementation had a general antioxidant effect in vivo in a natural aging model, which was also supported by improvements in aging-associated molecular parameters [4]. More specifically, the results of the above study showed that a daily intake of alkaline water supplementation (AWS) had a clear antiaging effect, as demonstrated by the increase in both telomerase activity and telomere length. The effects on telomeres were consistent with a significant reduction in ROS blood levels, with a mirror effect on both GSH and SOD-1, showing increased systemic levels. This was also consistent with an increased cellularity in both bone marrow and ovaries [4]. These results support a critical role of AWS in reducing oxidative stress in the whole body through a clear prevention of increased oxidation at the cellular and systemic levels.

The data of our study are supported by the results of some reports showing that water alkalinization counteracted the production and tissue accumulation of free radicals into the whole body [5] and improved survival in a model of long-lived mice, where the benefit of longevity was correlated with the consumption of alkaline water alone [6]. Moreover, water alkalinization by AWS was also shown to contribute to preventing prostate cancer [7] and to be included in the treatment of all cancers [8]. Additionally, the potential of the use of AWS in both preventing and curing human diseases is also supported by data showing that alkalinization may help in the control of gut microbiota [9] and in preventing and treating osteoporosis [10,11].

In the last decade, reports are increasingly supporting the use of hydrogen-rich water as a general antioxidant approach to prevent and, in some cases, cure diseases [12]. In fact, hydrogen-rich water was proven to be useful in preventing serious sickness in different physiological conditions, such as inflammatory responses [13], severe exercise in physically active males [14], and muscle soreness after resistance training [15]. However, it was shown that it may be helpful in treating a series of pathological conditions, such as liver injury [16], hypertension [17], chronic ulcerative colitis [18], type II diabetes [19], myocardial ischemia [20], graft-versus-host disease [21], and ureteral obstruction [22]. In addition, it may be helpful in preventing and treating potentially toxic conditions, such as radiotoxicity [23] and mercury toxicity [24].

Outside of the exciting pre-clinical evidence, hydrogen administration is problematic inasmuch as its inhalation is associated with a high risk of explosion, while the oral ingestion of hydrogen-saturated water may lead to clear biological advantages [25]. To overcome these problems, it has been shown that an injection of hydrogen-rich saline (HS) may have the same advantages while strongly reducing any possible danger. For example, HS could promote renal function recovery after ischemia/reperfusion injury in rats [26], exert a protective effect against cisplatin-induced ovarian injury [27], and protect neurological cells and fibers against different kinds of damages [28,29]. This may occur through different mechanisms, including an antioxidant effect. The antioxidant effect leads to the prevention of endoplasmic reticulum stress and the inhibition of autophagy. All this evidence has also been supported by a report showing that HS may reduce neurobehavioral deficits and apoptosis resulting from hypoxia/ischemia-induced brain damage [30].

For the above reasons, we decided to test water that was both hydrogen-rich and alkaline to restore and maintain a healthy state in a murine model. For this purpose, we administered a hydrogen-rich and alkaline water treatment orally to mice that had undergone a 2-week treatment with H_2_O_2_, which induced a general state of sickness related to a marked redox imbalance at both systemic and single-organ levels. The results show that the oral treatment with HAW reverted the H_2_O_2_ toxic effect while restoring the redox balance in treated mice.

## 2. Results

### 2.1. Systemic Response to the HAW Treatment

The whole set of experiments was aimed at verifying the ability of HAW to restore a general redox balance in mice treated with H_2_O_2_. For this purpose, mice were treated with H_2_O_2_ for two weeks and then drank either simple water only or HAW. Both groups were compared to mice that received water only from the beginning of the in vivo experiment. At the time of sacrifice, blood samples and organs were obtained from each mouse and analyzed for their redox balance and various cellular functions.

#### 2.1.1. The Antioxidant Effect of HAW

We first measured the plasmatic levels of both oxidant (i.e., RHO) and antioxidant (i.e., glutathione) molecules in both mice treated with H_2_O_2_ alone and those receiving HAW after the H_2_O_2_ treatment. The results show that the ROS plasmatic levels were significantly increased in the H_2_O_2_ group compared to the mice receiving only water, while the HAW treatment dramatically reduced the ROS plasmatic levels in the H_2_O_2_-treated mice (Figure 1a) (*p* < 0.001), thus restoring the condition of the mice receiving water only from the beginning of the experimental period (Figure 1a). The dramatic reduction of the ROS plasmatic levels was consistent with a significant increase in the glutathione (GSH) levels in the HAW-treated mice. The results also show that the HAW-treated mice had significantly higher GSH plasmatic levels compared to both the H_2_O_2_-treated mice and untreated controls (Figure 1b). Further analysis confirmed that the increased GSH plasmatic levels were significantly related to the decreased ROS levels in the HAW-treated mice (Figure 1b) (*p* < 0.001). This set of results strongly supports a systemic antioxidative effect of HAW, based on an increase in GSH plasmatic levels and re-established redox balance after a H_2_O_2_ treatment.

#### 2.1.2. DNA Damage

To assess the level of systemic DNA damage, we used a specific marker, 8-hydroxy-2′-deoxyguanosine (8-OHdG). 8-OHdG is the product of the interaction between oxygen-free radicals and the nucleobases of the DNA strand, such as guanine; it is commonly considered a DNA damage marker [31,32], and it is associated with carcinogenesis and degenerative diseases [33,34]. This set of experiments was aimed at evaluating the relationship between the HAW-induced systemic antioxidant response and the 8-OHdG serum levels. To this purpose, we measured the 8-OHdG serum concentrations in the absence or presence of HAW oral administration (Figure 2). The results show that the mice treated with HAW had significantly decreased serum levels of 8-OHdG (Figure 2) (*p* < 0.001) as compared to the mice exclusively treated with H_2_O_2_, reaching values lower than the control H_2_O group (1.22 ± 0.22 ng/mL, SD = ± 0.71 ng/mL). Comparing Figure 1 to Figure 2, it appears clear that the highest ROS levels corresponded to the highest systemic levels of DNA damage; while the highest GSH levels corresponded to the values of serum 8-OHdG comparable to those quantified in mice that did not receive H_2_O_2_ treatment, suggesting proper DNA repair related to the antioxidant response following HAW intake.

### 2.2. The Response at the Organ Level

Redox imbalance is involved in the regulation of many biological and physiological processes [35]. These include cell proliferation, the induction of apoptosis, cell senescence, and the regulation of the immune response [36,37]. We thus performed a series of analyses aimed at verifying the level of cellular fitness in the various organs of mice receiving either HAW or simple water after the H_2_O_2_ treatment. To this purpose, we decided to study the organs that provide the opportunity to obtain a single-cell suspension, thus allowing an ex vivo evaluation of different parameters (i.e., spleen and bone marrow). All the tests were performed with 1 × 10^6^ cells for each test.

#### 2.2.1. Oxidative Stress

Based on the above background in this set of experiments, we wanted to investigate the antioxidant effect of the HAW oral treatment in cell suspensions obtained from different organs of the mice (i.e., spleen and bone marrow). To this purpose, we obtained single and viable cell suspensions from the spleen and the bone marrow of the mice under the various treatment conditions.

We thus measured the ROS levels in cell suspensions isolated from the femoral bone marrow and the spleen of the mice. The results show that the H_2_O_2_ treatment induced a marked increase in the ROS levels in the cell suspensions from both the spleen (Figure 3a) and the bone marrow (Figure 3b) of the mice compared to the untreated mice receiving H_2_O only. The HAW treatment fully restored normal ROS levels in both the spleen (Figure 3a) and the bone marrow (Figure 3a) cell suspensions compared to the group receiving H_2_O_2_ only. Thus, treatment with HAW fully restored the redox balance in the immune organs of the H_2_O_2_-treated mice.

#### 2.2.2. Oxidative Stress in Mitochondria of Bone Marrow Cells and Splenocytes

In this set of experiments, we wanted to correlate the systemic and organ antioxidant response to the mitochondrial damage in the spleen and the bone marrow obtained from the mice treated or not with HAW following two weeks of H_2_O_2_ intake. We found that H_2_O_2_ induced an increase in ROS at both the systemic and organ levels and that HAW intake entirely restored the redox balance by reducing the ROS levels (Figure 1 and Figure 3). Here, we evaluated the mitochondrial stress through the measurement of both the mitochondrial membrane potential and the mitochondrial superoxide. The results first show that the H_2_O_2_ treatment induced an increase in the mitochondrial membrane potential in both the splenocytes (Figure 4a) and the bone marrow cells (Figure 4b) compared to the mice that received H_2_O only. The splenocytes and bone marrow cells from the mice that orally received HAW after H_2_O_2_ treatment showed significantly lower ΔΨm in comparison to the H_2_O_2_ group, leading to a condition entirely comparable to that observed in the mice that received plain H_2_O from the beginning of the experiment (i.e., a physiological condition).

Mitochondrial superoxide is a highly reactive molecule that is usually converted in water and oxygen by the cellular antioxidant system, often overcoming the cellular antioxidant pathways and leading to protein, lipid, and DNA damage [38,39]. For the above reasons, it is considered a major cause of cellular damage induced by oxidative stress. We measured the mitochondrial superoxide in the same cells comparing the various mice groups. The results show that the mitochondrial superoxide levels were significantly higher in the mice treated with H_2_O_2_ compared to the mice who received H_2_O only, in both their splenocytes (Figure 5a) and bone marrow (Figure 5b). Again, the HAW oral treatment induced a significant reduction of mitochondrial superoxide (in both splenocytes (Figure 5a) and bone marrow (Figure 5b), reaching values comparable to the mice drinking plain H_2_O from the beginning of the experiment.

This set of experiments showed that the HAW oral treatment, together with restoring the ROS levels at both the systemic and organ levels, led to normalization of both the mitochondrial membrane potential and the mitochondrial superoxide levels induced by H_2_O_2_ treatment, reaching levels comparable to the control mice and suggesting that HAW oral treatment could be helpful in counteracting aging.

#### 2.2.3. Ex Vivo Proliferative Response of Single-Cell Suspensions from Spleen and Bone Marrow

In order to support the results obtained in the redox balance analysis, this set of experiments was aimed at assessing the spontaneous proliferative response of both splenocytes and bone marrow cells. To this purpose, we analyzed the spontaneous proliferation in cells isolated from the bone marrow and spleen cells of mice that received HAW following two weeks of H_2_O_2_ treatment compared to the mice left untreated after the H_2_O_2_ period and the mice that received plain water only from the beginning of the experiment. Consistent with all the other analyses, the H_2_O_2_ oxidative stress induced a marked and significant decrease in the proliferation in both bone marrow (*p* < 0.05, Figure 6a) and splenocytes (*p* < 0.001, Figure 6b) compared to the group that exclusively received H_2_O.

The group of mice receiving HAW after the H_2_O_2_ treatment showed significantly higher spontaneous proliferation in both the bone marrow cells (*p* < 0.05, Figure 6a) and the splenocytes (*p* < 0.05, Figure 6b) compared to the H_2_O_2_ group. It is interesting to note that the HAW treatment brought the values of spontaneous proliferation to those observed in the mice that did not receive H_2_O_2_ in both the bone marrow cells and splenocytes (Figure 6). This set of results shows how HAW treatment is able to restore proper spontaneous proliferation in organs that are key in the immune response, such as the spleen and bone marrow.

## 3. Discussion

In this study, we investigated the effects of hydrogen-rich and alkaline water (HAW) on the redox balance in a model of mice in which redox imbalance was induced through a daily intake of H_2_O_2_.

Previous data have shown that hydrogen-rich water may either prevent or cure many diseases [12,13]. Hydrogen-rich water may have a prevention effect on both inflammatory reactions and side effects of severe physical exercise [15], including resistance training [14]; it may also prevent or cure potentially toxic conditions, such as radiotoxicity [16] and mercury toxicity [24]. However, it may be helpful in curing some pathological conditions, such as liver injury [16], hypertension [17], chronic ulcerative colitis [18], type II diabetes [19], myocardial ischemia [20], graft-versus-host disease [21], and ureteral obstruction [22].

In a previous study, we showed that alkaline water supplementation induced a clear systemic antioxidant effect in a natural aging model, demonstrated by the decreased ROS counteracted by the increased GSH and SOD-1 plasmatic levels. This was consistent with an antiaging and antioxidant effect at the organ level [4], thus supporting a role of alkaline water in reducing the oxidative stress in the whole body and at the cellular level, leading to a slower aging process. In supporting these data, additional reports have shown that alkaline water had a “sparing effect” on antioxidant enzyme levels [5] and improved the survival in a model of long-lived mice [6]. Moreover, water alkalinization was proven useful in both preventing [7] and treating cancers when combined with chemotherapy [8]. Alkalinization may also have a part in the control of gut microbiota [9] and in preventing osteoporosis [10,11].

Together, these data strongly support a critical role of alkaline water in both the prevention and cure of human diseases.

Here, we have presented a commercially available new compound, represented by a hydrogen-rich alkaline water (HAW), with which we treated mice in which a serious redox imbalance was induced by a 2-week administration of H_2_O_2_-supplemented water. The results show that the oral treatment with HAW dramatically restored the redox balance in the H_2_O_2_-treated mice. This effect was demonstrated by a systemic reduction of ROS levels, consistent with increased plasmatic levels of the antioxidant glutatione.

This general antioxidant response was also consistent with reduced DNA damage, shown by the significant reduction of the circulating levels of 8-hydroxy-2′-deoxyguanosine (8-OHdG). Our results show that HAW reduced the plasmatic levels of 8-OHdG, supporting the evidence of reduced DNA damage throughout the body. Comparing the two sets of results, it is clear that the highest ROS levels corresponded to increased DNA damage, while the highest GSH levels corresponded to reduced DNA damage, suggesting proper DNA repair related to the antioxidant response following HAW intake.

To further support these findings, we explored the level of redox balance in the organs of the various groups of mice. To this purpose, we investigated the effect of HAW oral treatment on oxidative stress in cell suspensions obtained from organs involved in the murine immune response (i.e., spleen and bone marrow). Cell suspensions from both the spleen and bone marrow always showed reduced ROS levels when obtained from mice that receive HAW. This finding shows that the administration of HAW fully restored the redox balance in the immune organs of H_2_O_2_-treated mice.

In this study, we also measured the mitochondrial membrane potential (ΔΨm), which is directly related to redox imbalance [40,41]. ΔΨm is a dynamic parameter that varies depending on the state of activation/differentiation of the cell; its stable alteration may lead to cell death [40,42,43]. Lastly, scientific evidence has shown that a stable increase in ΔΨm induces a drastic increase in ROS levels at both the systemic and organ levels [43,44,45,46]. For the above reasons, we investigated the mitochondrial damage in the spleen and the bone marrow obtained from the mice treated or not with HAW following two weeks of the H_2_O_2_ intake. We measured both the mitochondrial membrane potential and the mitochondrial superoxide in single-cell suspensions obtained from the spleen and bone marrow of mice in the various in vivo conditions. The H_2_O_2_ treatment induced an increase in the mitochondrial membrane potential in both the splenocytes and the bone marrow cells, while HAW intake induced lower ΔΨm, thus restoring physiological conditions. We also measured the levels of mitochondrial superoxide and highly toxic molecules related to cell damage and redox imbalance [47]. The results show that the mitochondrial superoxide levels were significantly higher in both splenocytes and bone marrow cells of the mice treated with H_2_O_2_, while the HAW oral treatment induced a significant reduction of the mitochondrial superoxide levels in both the splenocytes and bone marrow cells. These results show that the HAW oral treatment, together with restoring the ROS levels at both the systemic and organ levels, led to the normalization of both the mitochondrial membrane potential and the mitochondrial superoxide levels induced by H_2_O_2_ treatment, reaching levels comparable to the control mice. We used a model that heavily affects the body homeostasis [48], and here, we have shown that HAW was able to entirely revert this unbiased condition. The last result we obtained in this study was on spontaneous cell proliferation. The H_2_O_2_ treatment dramatically reduced the spontaneous proliferation in both the spleen and bone marrow cell suspensions. HAW administration after two weeks of H_2_O_2_ entirely restored the spontaneous proliferation, leading to values similar to those measurable in mice that did not receive H_2_O_2_. This is a crucial result supporting the general healthy effect of HAW also in conditions of heavy redox imbalance. Figure 7 shows the in vivo model used in this study and summarizes the results obtained with the simple addition of hydrogen-rich/alkaline water to the daily intake of water.

All in all, our data suggest that the intake of HAW may efficiently counteract aging by inducing a systemic antioxidant response. Additionally, the results obtained in a model where heavy redox imbalance was induced support the use of hydrogen-rich and alkaline water in counteracting a series of systemic toxic conditions in which severe body over-oxidation occurs. These conditions may include metabolic syndromes, tumors, poisoning, chronic exposure to heavy metals, chronic inflammation, and neurological diseases.

## 4. Materials and Methods

### 4.1. In Vivo Studies

C57BL/6J female mice between 16 and 20 g (4 weeks of age) were purchased from Charles River Laboratories Italia s.r.l., (Calco, Lecco, Italy) and housed in the animal facility of the Italian National Institute of Health. The mice had 10 and 14 h periods of light and darkness, respectively, and were housed in different numbers of animal cages, depending on the experiment, with ad libitum mice chow (Mucedola, Settimo Milanese (MI), Italy) and water provided through a bottle. The mice were checked twice a week by a veterinarian responsible for animal welfare monitoring for signs of sufferance, such as weight loss, decreased water and food consumption, poor hair coat, decreased activity levels, according to the guidelines for correct laboratory practice and signs of poor quality of life.

The mice were divided into 3 groups, with 10 mice/group: the control group received sterilized tap water only (H_2_O group), one group was treated with hydrogen peroxide (H_2_O_2_ group), and one group was pretreated with hydrogen peroxide and then treated with HAW (HAW group). The H_2_O group was not treated and drank only untreated water; the H_2_O_2_ group was treated with 1% hydrogen peroxide (H_2_O_2_) dissolved in water for two weeks and then given untreated water; the HAW group was treated with 1% hydrogen peroxide (H_2_O_2_) dissolved in water for two weeks, and then given untreated water or HAW for three weeks. Just before sacrificing the mice, blood was drawn from their eyes. Immediately after the sacrifice, bone marrow was isolated from both the tibias and femurs of the hind legs, while splenocytes were obtained from the spleen. Blood, bone marrow cells, and splenocytes were used for the subsequent experimental analyses of aging and antioxidant parameters.

### 4.2. Collection and Processing of Murine Plasma from Blood Samples

Blood sample collection from the CTR and HAW mice was performed by retro-orbital bleeding (ROB) immediately before the sacrifice. This safe phlebotomy technique allowed for obtaining high-quality samples of an adequate volume (500 µL/mouse) for further analysis. A total of 52 blood samples were collected in K3-EDTA-coated collection tubes. To obtain the plasma samples, EDTA-treated whole blood from both the CTR and AWS groups was centrifuged at 400× *g* for 20 min. The plasma samples (250 µL/mouse) were then collected and immediately analyzed or stored at −80 °C until analysis.

### 4.3. Bone Marrow and Splenocytes Cells Recovery from C57BL/6J Mice

Immediately after the sacrifice of the CTR and HAW mice, their spleen and bone marrow (from both the tibias and femurs of the hind legs) were obtained and placed in physiological solution (NaCl). The bone marrow and the spleen were disrupted with the blunt end of a 5 mL syringe plunger. Both the splenocytes and bone marrow cells were isolated using a Falcon^®^ 100 µm cell strainer (Corning, NY, USA), obtaining a uniform single-cell suspension from the bone marrow. The single-cell suspensions were washed twice in PBS and immediately processed for the following analysis.

### 4.4. Hydrogen-Rich Alkaline Water Supplementation (HAW)

HAW consists of commercially available powder called pHLife^®^ (by Vivere Alcalino, Brindisi, Italy). pHLife^®^ supplement powder consists of potassium carbonate, calcium ascorbate dihydrate (E302), sodium ascorbate (E301), sodium chloride, ascorbic acid (Vitamin C), potassium anhydrous sulfate, anhydrous carbonate sodium (E500(i)), magnesium oxide, chromium picolinate, sodium selenite, citric acid. pHLife, is currently available for humans in Italy and consumed by thousands of people yearly.

HAW was obtained by adding 1 g of pHLife powder to 1 L of tap water, until reaching pH 9.0.

Hydrogen enrichment was achieved through a patented procedure (patent No. 102018000006414). Briefly, the above detailed pHLife^®^ formulation was added to water at a dosage of 1 g/L. Thanks to its particular blend of minerals and vitamin C, the contact with water produces molecular hydrogen at a concentration of 1 ppm per liter, as measured by the TRUSTLEX ENH-1000 Portable Dissolved Hydrogen Meter (Suita, Japan). Production of molecular hydrogen occurs with the breaking of the hydrogen bond present in the molecule of ascorbic acid (C_6_H_8_O_6_) and calcium ascorbate (C_12_H_14_CaO_12_) in contact with water and CO_2_. The hydrogen atoms released in the reaction recombine into single (H_2_) molecules and H_2_O. H_2_ gas remains for various minerals in hermetically sealed containers and for some minerals in containers in contact with air.

HAW was administered orally to the mice at a volume of 250 mL/kg/mouse (5 mL/mouse) every day without interruption, corresponding to a concentration of 215 mg/kg/mouse/day (3.6 µL/mouse/day) of HAW. The daily treatment of the mice with HAW started 2 weeks after the mice arrived at the animal facility (at 6 weeks of age) and continued for 10 months until the sacrifice of the mice (at 51 weeks of age).

The daily intake of each supplemented element is shown in Table 1, the HAW composition is shown in Table 2.

### 4.5. Total ROS Assay

The plasma samples underwent analysis using a Total Reactive Oxygen Species (ROS) Assay Kit 520 nm (ThermoFisher, Waltham, MA, USA). In detail, the plasma samples were incubated with 100 µL of 1X ROS Assay Stain for 60 min in a 37 °C incubator with 5% CO_2_. The samples were treated with the desired reagents to induce the production of ROS and analyzed on a microplate reader off the 488 nm (blue laser) in the FITC channel.

### 4.6. GSH Detection and Quantification Assay

The plasma samples underwent analysis using a colorimetric assay, the Glutathione Colorimetric Detection Kit (ThermoFisher, USA). The samples were incubated for 20 min at room temperature after the addition of the detection reagent and reaction mixture. The optical densities were recorded at 405 nm.

### 4.7. Mitochondrial Membrane Potential Measurement

MitoTracker^®^ Mitochondrion-Selective Probes (Molecular Probes, Invitrogen, Thermo Fisher Scientific, Waltham, MA, USA) were used to measure the mitochondrial membrane potential in the bone marrow cells and splenocytes of the mice, which were isolated right after the mice were sacrificed. These probes are a green-fluorescent stain that seems to target mitochondria irrespective of their membrane potential. To mark the mitochondria, live cells were treated with 100 nM of MitoTracker^®^ probes. These probes diffused passively across the plasma membrane and gathered in active mitochondria. The probes, in their reduced state, do not fluoresce until they enter live cells. Once inside, they are oxidized to the fluorescent mitochondrion-selective probe and become trapped in the mitochondria. The suspension cells were centrifuged to obtain a cell pellet, which was then resuspended in a MitoTracker^®^ probe staining solution that had been prewarmed to 37 °C. After incubating for 30 min at 37 °C and 5% CO_2_, the cells were centrifuged again to re-pellet them and then resuspended in a fresh prewarmed medium or buffer. The green fluorescence was measured at Ex/Em = 490/516 nm using a microplate reader. The data are reported as the mean fluorescence intensity (M.F.I.) in arbitrary units (a.u.).

### 4.8. Mitochondrial Superoxide Detection

The presence of mitochondrial superoxide in the bone marrow cells and splenocytes of mice (which were immediately isolated post-sacrifice) was determined using a Mitochondrial Superoxide Detection Kit (supplied by Abcam, based in Cambridge, UK). This kit employs a sensitive, one-step fluorometric assay that utilizes MitoROS 580 dye to detect the intracellular superoxide radical in live cells. The dye, which can permeate cells, selectively interacts with mitochondrial superoxide in live cells, resulting in the production of a red fluorescence signal. This signal was measured at Ex/Em = 540/590 nm using a microplate reader following a 60 min incubation at 37 °C. The results were calculated from the difference in the fluorescence intensity between the control and treated cells, and are reported as the mean fluorescence intensity (M.F.I.) in arbitrary units (a.u.).

### 4.9. Cell Proliferation Assay

The proliferation of bone marrow cells and splenocytes was measured using Alkaline Phosphatase Yellow (pNPP) Liquid Substrate for ELISA (Sigma-Aldrich, St. Louis, MO, USA). This product is supplied as a ready-to-use buffered alkaline phosphatase substrate containing p-nitrophenylphosphate (pNPP). Following the manufacturer’s protocols, the solution (100 µL/well) was added to cells seeded in a 96-well plate. After the reaction with alkaline phosphatase, a yellow reaction product was formed and read at 405 nm using a microplate reader (BioTek Epoch, Agilent, Santa Clara, CA, USA). The data are expressed as the optical density × 1000 (arbitrary units, a.u.).

### 4.10. Statistical Analysis

The results in the text are expressed as the means ± standard error (SE), calculated using GraphPad Prism software 9 (San Diego, CA, USA). The statistical analysis was performed with an unpaired *t*-test (Student’s *t*-test). Statistical significance was set at *p* < 0.05.

## Figures and Tables

**Figure 1 ijms-25-06736-f001:**
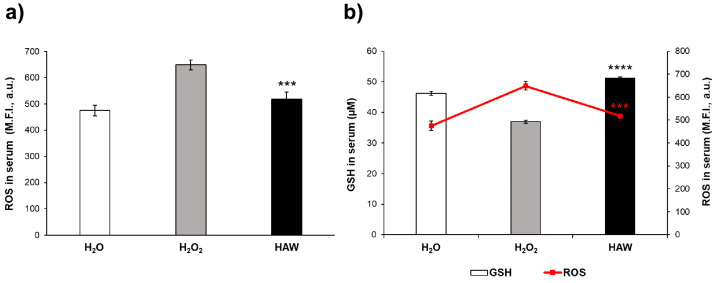
Antioxidant effect of HAW in C57BL/6J female mice. (**a**) Serum ROS levels in C57BL mice after treatment with HAW. Analysis of the total ROS levels (mean fluorescent intensity (M.F.I.); see Section 4) was performed in the serum samples collected just before the sacrifice of the untreated mice (white histogram), those receiving H_2_O_2_ only (gray histogram), and those treated with HAW (black histogram). (**b**) GSH serum levels in the various mice’ groups compared to the ROS plasmatic levels in the same animals (µM). Data are expressed as the means ± SE. *** *p* < 0.001, **** *p* < 0.0001.

**Figure 2 ijms-25-06736-f002:**
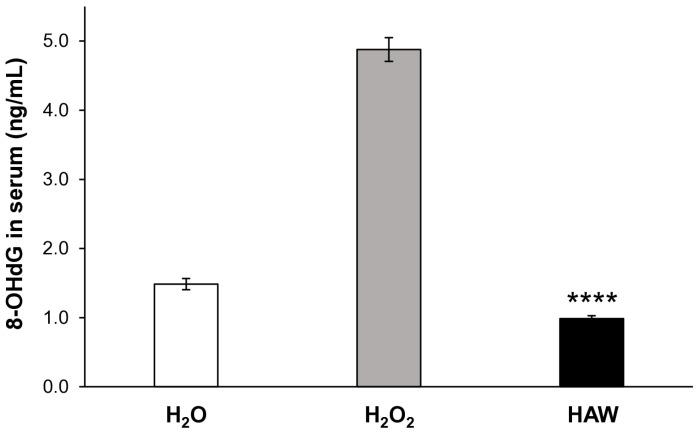
8-OHdG serum levels in untreated mice (white histogram), mice receiving H_2_O_2_ only (gray histogram), and mice treated with HAW (black histogram). The absorbance was read at 450 nm using a microplate reader, and the concentration of 8-OHdG (ng/mL) was calculated from the standard curve. Data are expressed as the means ± SE. **** *p* < 0.0001.

**Figure 3 ijms-25-06736-f003:**
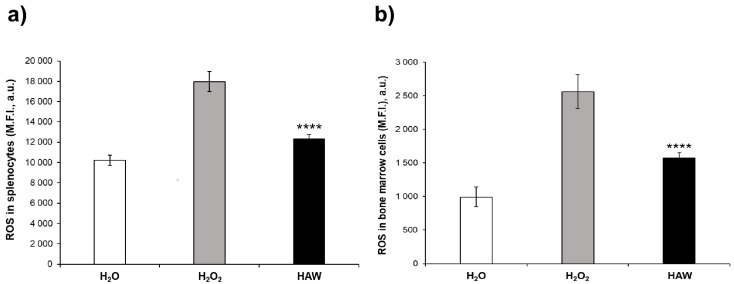
Effects of HAW treatment on ROS levels in splenocytes and bone marrow cells obtained from untreated mice (white histogram), those receiving H_2_O_2_ only (gray histogram), and those treated with HAW (black histogram). (**a**) ROS levels (M.F.I., a.u.) in splenocytes measured using a fluorescence microplate reader at 488 nm (blue laser). (**b**) ROS levels (M.F.I., a.u.) in bone marrow cells measured using a fluorescence microplate reader at 488 nm (blue laser). Data are expressed as the means ± SE. **** *p* < 0.0001.

**Figure 4 ijms-25-06736-f004:**
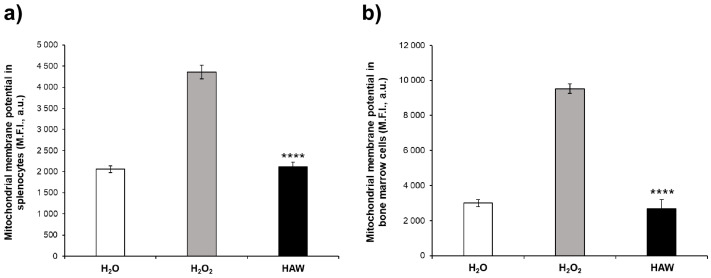
Effect of HAW treatment on mitochondrial membrane potential in splenocytes and bone marrow cells obtained from mice either untreated (white histogram) or receiving H_2_O_2_ only (gray histogram) or treated with HAW (black histogram). (**a**) Mitochondrial membrane potential (M.F.I., a.u.) measurement in splenocytes. Green fluorescence values were read at Ex/Em = 490/516 nm in a fluorescence microplate reader. (**b**) Mitochondrial membrane potential (M.F.I., a.u.) measurement in bone marrow cells. Green fluorescence values were read at Ex/Em = 490/516 nm using a fluorescence microplate reader. Data are expressed as the means ± SE. **** *p* < 0.0001.

**Figure 5 ijms-25-06736-f005:**
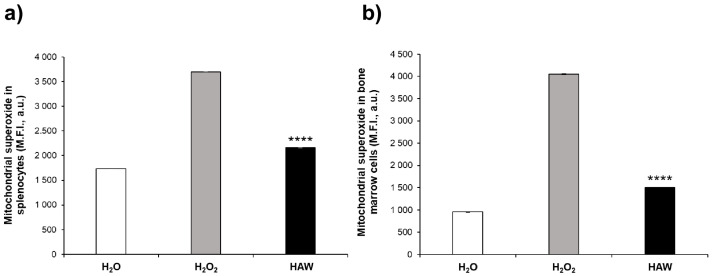
Effect of HAW treatment on mitochondrial superoxide in splenocytes and bone marrow cells obtained from untreated mice (white histogram), those receiving H_2_O_2_ only (gray histogram), and those treated with HAW (black histogram). (**a**) Mitochondrial superoxide (M.F.I., a.u.) measurement in splenocytes. Red fluorescence values were read at Ex/Em = 540/590 nm using a fluorescence microplate reader. (**b**) Mitochondrial superoxide (M.F.I., a.u.) measurement in bone marrow cells. Red fluorescence values were read at Ex/Em = 540/590 nm using a fluorescence microplate reader. Data are expressed as the means ± SE. **** *p* < 0.0001.

**Figure 6 ijms-25-06736-f006:**
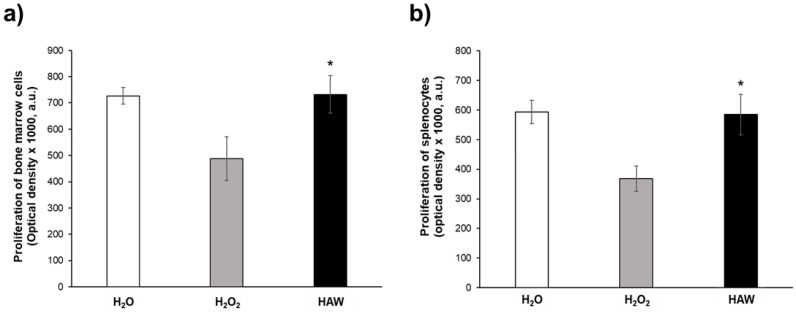
Effect of HAW treatment on proliferation of bone marrow cells and splenocytes obtained from untreated mice (white histogram), those receiving H_2_O_2_ only (gray histogram), and those treated with HAW (black histogram). The optical density values (a.u.) were read at 405 nm using a microplate reader after the reaction with alkaline phosphatase. (**a**) Proliferation of bone marrow cells. (**b**) Proliferation of splenocytes. Data are expressed as the means ± SE. * *p* < 0.05.

**Figure 7 ijms-25-06736-f007:**
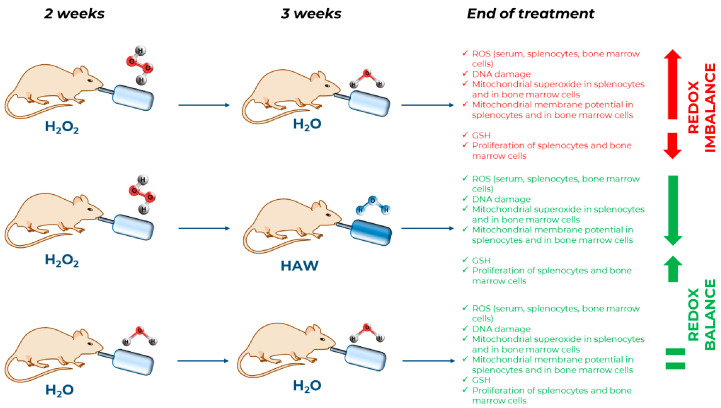
Experimental scheme of treatment of C57BL/6J mice with HAW following oxidative stress. HAW treatment restores the physiological oxidant–antioxidant balance in mice.

**Table 1 ijms-25-06736-t001:** Information on the single HAW components.

SUBSTANCE	DOSE FOR 1 CPS	MAX DAILY DOSE	%Nutrient Reference Values (NRV)
Potassium	250.3 mg	751.1 mg	37.56%
Sodium	79.34 mg	238.02 mg	-
Vitamin C	333.33 mg	1000 mg	1250%
Chloride	60.9 mg	182.72 mg	22.84%
Chromium	6.66 mcg	20 mcg	50%
Selenium	9.16 mcg	27.5 mcg	50%

**Table 2 ijms-25-06736-t002:** HAW composition.

Salt	Source	Powder %
Sodium Chloride (NaCl)	Bolivian RoseTM—Andes Mountain Salt from Andes Mountain Range (Bolivia) (Woodinville, WA, USA)	95.5
	Natural Alps Mountains Salt, coarse from Bergkern (Austria)	94.5
	HALITE SALT from rock salt mines (Pakistan)	99.35
Sodium (Na)	Bolivian RoseTM—Andes Mountain Salt from Andes Mountain Range (Bolivia) (Woodinville, WA, USA)	/
	Natural Alps Mountains Salt, coarse from Bergkern (Austria)	35.4
	HALITE SALT from rock salt mines (Pakistan)	/
Calcium (Ca)	Bolivian RoseTM—Andes Mountain Salt from Andes Mountain Range (Bolivia) (Woodinville, WA, USA)	0.7
	Natural Alps Mountains Salt, coarse from Bergkern (Austria)	0.25
	HALITE SALT from rock salt mines (Pakistan)	0.19
Magnesium (Ma)	Bolivian RoseTM—Andes Mountain Salt from Andes Mountain Range (Bolivia) (Woodinville, WA, USA)	0.208
	Natural Alps Mountains Salt, coarse from Bergkern (Austria)	0.1
	HALITE SALT from rock salt mines (Pakistan)	0.16
Potassium (K)	Bolivian RoseTM—Andes Mountain Salt from Andes Mountain Range (Bolivia) (Woodinville, WA, USA)	0.646
	Natural Alps Mountains Salt, coarse from Bergkern (Austria)	0.3
	HALITE SALT from rock salt mines (Pakistan)	/
Iron (Fe)	Bolivian RoseTM—Andes Mountain Salt from Andes Mountain Range (Bolivia) (Woodinville, WA, USA)	0.303
	Natural Alps Mountains Salt, coarse from Bergkern (Austria)	/
	HALITE SALT from rock salt mines (Pakistan)	/
Sulphate (SO_4_)	Bolivian RoseTM—Andes Mountain Salt from Andes Mountain Range (Bolivia) (Woodinville, WA, USA)	/
	Natural Alps Mountains Salt, coarse from Bergkern (Austria)	2.0
	HALITE SALT from rock salt mines (Pakistan)	0.13

## Data Availability

The data presented in this study are available on request from the corresponding author.

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
