# Peer review of "Hydrogen-Rich Alkaline Water Supplementation Restores a Healthy State and Redox Balance in H2O2-Treated Mice"

_ijms, 2024, doi:10.3390/ijms25126736_

Round 1

Reviewer 1 Report

Comments and Suggestions for Authors

Dear authors, thank you for very interesting article. HAW seems to be very promising chemical/solution with an antioxidant activity.

I have some comments and questions:

Comment 1: Figures (no. 1;3;4;and 5) should be smaller and next to each other - their locations should be A on the left side and B on the right side. Now you have an A-part figure on the top and B-part bellow it.

Comment2: Please add more info about product pHLife-producer in the part Materials and Methods. And more info about this product should be also given to part introduction. This commercial product (capsules) is freely available for people. 

Question1: Is safe to drink HAW by humans? In connection with previous comments, pHLife capsules are available for people. But you have not added any data about the previous test results of this product. It should be added.

I suppose that that this product was tested before it was put on the market.

Author Response

REVIEWER 1

Dear authors, thank you for very interesting article. HAW seems to be very promising chemical/solution with an antioxidant activity.

We thank you the reviewer indeed

I have some comments and questions:

Comment 1: Figures (no. 1;3;4;and 5) should be smaller and next to each other - their locations should be A on the left side and B on the right side. Now you have an A-part figure on the top and B-part bellow it.

Figures have been modified accordingly with the reviewer’s suggestion

Comment2: Please add more info about product pHLife-producer in the part Materials and Methods. And more info about this product should be also given to part introduction. This commercial product (capsules) is freely available for people. 

More details on the products are now included in the required sections

Question1: Is safe to drink HAW by humans? In connection with previous comments, pHLife capsules are available for people. But you have not added any data about the previous test results of this product. It should be added.

I suppose that that this product was tested before it was put on the market.

HAW, pHLife, is currently available for humans and many people are assuming it. being a supplier, containing salts and vitamin C. The pHLife’s ingredients have been all released by the european regulatory agency for food suppliers, as it is clear from the product’s label, (while in italian).   Moreover, it is notified in the Italian Ministry of Health’s register of food suppliers.  pHLife is present in the italian market since 2020: From that date it is assumed by thousands of people without any evidence of adverse events, while only positive effects on the general health of the body.   The flyer below is included for the reviewer’s use only

However, together with the absence of side effects in humans, this study witnesses its safety in animal models

We sincerely thank the reviewer for her/his suggestions that helped us to improve the quality of our manuscript

Reviewer 2 Report

Comments and Suggestions for Authors

In the present paper Mizzoni and collegues investigated the anti-oxidant potential of hydrogen-rich alkaline water (HAW) in a mouse model of induced oxidant imbalance due to H2O2 administration. The authors investigated the role of HAW in hampering systemic oxidative stress by measuring the circulating levels of ROS, 8-oxo-dG, glutathione and mitochondrial membrane potential. Moreover, the red-ox status of bone marrow and spleen cell suspensions from mice groups were investigated. The authors speculated the healthy effect of HAW in counteracting red-ox imbalance.

Although the topic is of interest, some issues need to be addressed before considering the paper suitable for publication.

MAJOR ISSUES

1. The manuscript shows some imperfections. The material and method paragraph need to be strongly revised. The authors reported sub-paragraphs containing the same title but reporting different information. Moreover, they referred to ovarian cell analysis that was missing in the results as well as did not provide indication about 8-oxo-dG evaluation.

2. In my opinion, it is not clear why the authors looked only at bone marrow and spleen to assay the organ specific effects induced by H2O2 treatment and counteracted by HAW. Given the route of exposure, it would be more indicated if the authors could investigate more specific organs, such as liver, kidney and cardiovascular system.

3. As concern the cell suspensions used for ROS and mitochondrial membrane potential, the authors must indicate the cell content of the suspension used for the quantitative assays. I obviously assume that it is the same for all the samples, but a clear indication must be reported within the text.

4. A detailed analysis of the red-ox imbalance on other key aspects of cell biology needs to be done. For example, the expression levels of SOD, Catalase, their ratio and the rate of lipid peroxidation must be analyzed. Moreover, the analysis of the rate of apoptosis and senescence are needed to provide indications of the protective role of HAW in counteracting red-ox induced damages (see PMID: 19804370; PMID: 38700499).

5. A figure/scheme summarizing the experimental plan could be useful for easily understanding the rationale of the study.

MINOR ISSUES

Table 1: please explicit the title of the 4th column; %VNR is poor clear.

Table 2: what does the % indicated in title of column 3 stand for? Please modify.

Line 75: “HI injury”, please explicit the acronym HI, this is the first time apparency in the text

Comments on the Quality of English Language

 Minor check of the english 

Author Response

REVIEWER 2

In the present paper Mizzoni and collegues investigated the anti-oxidant potential of hydrogen-rich alkaline water (HAW) in a mouse model of induced oxidant imbalance due to H2O2 administration. The authors investigated the role of HAW in hampering systemic oxidative stress by measuring the circulating levels of ROS, 8-oxo-dG, glutathione and mitochondrial membrane potential. Moreover, the red-ox status of bone marrow and spleen cell suspensions from mice groups were investigated. The authors speculated the healthy effect of HAW in counteracting red-ox imbalance.

Although the topic is of interest, some issues need to be addressed before considering the paper suitable for publication.

MAJOR ISSUES

  1. The manuscript shows some imperfections. The material and method paragraph need to be strongly revised. The authors reported sub-paragraphs containing the same title but reporting different information. Moreover, they referred to ovarian cell analysis that was missing in the results as well as did not provide indication about 8-oxo-dG evaluation.

The manuscript has been carefully revised for any imperfection, repetition and misinterpretation. Actually, the data on ovarian cells have been included in the early versions but eliminated in the final version inasmuch as it was misleading. Now ovarian cells have been eliminated from this paper. We sincerely apologize for the many errors and repetitions mostly due to the fact that it has been submitted in the meanwhile our laboratory was moving elsewhere.

  1. In my opinion, it is not clear why the authors looked only at bone marrow and spleen to assay the organ specific effects induced by H2O2treatment and counteracted by HAW. Given the route of exposure, it would be more indicated if the authors could investigate more specific organs, such as liver, kidney and cardiovascular system.

We decided to investigate spleen and bone marrow inasmuch as from those organs it is possible to obtain a single cell suspensions to study in ex vivo experiments. Liver, kidney and hearth do not allow to obtain a representative single cell suspension. We provided data on the blood in the general  assessment of the animal well being, in order to provide a reliable picture on the general health of the single animals.

  1. As concern the cell suspensions used for ROS and mitochondrial membrane potential, the authors must indicate the cell content of the suspension used for the quantitative assays. I obviously assume that it is the same for all the samples, but a clear indication must be reported within the text.

 We have added clearer information on the number of cells undergone ROS and mitochondrial membrane potential measurements. Actually, we used a minimum of 1 x 106 cells for each test.

  1. A detailed analysis of the redox imbalance on other key aspects of cell biology needs to be done. For example, the expression levels of SOD, Catalase, their ratio and the rate of lipid peroxidation must be analyzed. Moreover, the analysis of the rate of apoptosis and senescence are needed to provide indications of the protective role of HAW in counteracting red-ox induced damages (see PMID: 19804370; PMID: 38700499).

We thank the reviewer for her/its comments. Actually, we designed this study to evaluate the general effect of HAW on the health status of the mice following the H2O2 treatment. In this check up we have included the level of redox imbalance as evaluated by the systemic levels of both ROS and glutathione, as an example of a well-known anti-oxidant molecule. We wanted to emphasize the effect of HAW on the mitochondrial status as assessed by measurements of both mitochondrial membrane potential and mitochondrial superoxide in both splenocytes and bone marrow single cell suspensions, as a real picture of redox balance in the organs of HAW treated and untreated mice. The results are really clear in showing that HAW induced a real good fitness at the mitochondrial levels in the examined organs. We have newly included the level of proliferation in single cell suspensions of spleen and bone marrow cells in order to widen the evidence supporting the real effect of HAW in H2O2 treated mice and a new figure is included (Figure 6).

  1. A figure/scheme summarizing the experimental plan could be useful for easily understanding the rationale of the study.

A new figure (Figure 7) has been included in the revision showing both the experimental model and a summary of the results

MINOR ISSUES

Table 1: please explicit the title of the 4th column; %VNR is poor clear.

Table 2: what does the % indicated in title of column 3 stand for? Please modify.

Line 75: “HI injury”, please explicit the acronym HI, this is the first time apparency in the text

% VNR was the acronyme of the Italian statement corresponding to Nutritional Reference Values, and this now amended into the table

The % in column 3 is the % of the single component in the final powder

HI is the acronyme of Hypoxia/Ischemia

All these minor errors have been amended

The  english has been revised carefully all along the manuscript

We sincerely thank the reviewer for her/his suggestions that helped us to improve the quality of our manuscript

Round 2

Reviewer 2 Report

Comments and Suggestions for Authors

Lines 221-223: Much more experiments investigating the biology of ex vivo cells are needed to reach the speculation stated in lines 221-223. Please, change the sentence in a more appropriate way.

Author Response

we have changed the last sentence of the results (lines 221-223) as follows "This set of results show how HAW treatment is able to restore a proper spontaneous proliferation in organs that are key in the immune response, such as spleen and bone marrow".